# Pariah moonshine

John F.R. Duncan [1], Michael H. Mertens[2] & Ken Ono [1]

Finite simple groups are the building blocks of finite symmetry. The effort to classify them precipitated the discovery of new examples, including the monster, and six pariah groups which do not belong to any of the natural families, and are not involved in the monster. It also precipitated monstrous moonshine, which is an appearance of monster symmetry in number theory that catalysed developments in mathematics and physics. Forty years ago the pioneers of moonshine asked if there is anything similar for pariahs. Here we report on a solution to this problem that reveals the O'Nan pariah group as a source of hidden symmetry in quadratic forms and elliptic curves. Using this we prove congruences for class numbers, and Selmer groups and Tate–Shafarevich groups of elliptic curves. This demonstrates that pariah groups play a role in some of the deepest problems in mathematics, and represents an appearance of pariah groups in nature.

[1] Department of Mathematics and Computer Science, Emory University, 400 Dowman Drive, Atlanta, GA 30322, USA. [2] Mathematisches Institut der Universität zu Köln, Weyertal 86-90, D-50931 Köln, Germany. Correspondence and requests for materials should be addressed to J.F.R.D. (email: john.duncan@emory.edu) or to M.H.M. (email: mmertens@math.uni-koeln.de) or to K.O. (email: ono@mathcs.emory.edu)

As atoms are the constituents of molecules, the finite simple groups are the building blocks of finite symmetry. The question of what finite simple groups are possible was posed[1] in 1892. By the 1950s it was expected that most should belong to certain infinite families which are naturally defined in geometric terms. For example, the rotational symmetry of a regular polygon with a prime number of edges—a cyclic group of prime order—is a finite simple group. The next example is the rotational symmetry of a regular dodecahedron, but to see it as part of a family it should be regarded differently, via its action on five embedded tetrahedrons, for instance.

In 1963 the monumental Feit–Thompson odd order paper[2] established that any non-cyclic finite simple group must have a two-fold symmetry inside. This led to a surge of activity in group theory which, despite uncovering unexpected examples, paved the way for Gorenstein's 1972 proposal[3] to classify finite simple groups completely. Building upon thousands of pages of published papers, this program was completed[4, 5] in 2002. The resulting classification is a crowning achievement of twentieth century mathematics. Thompson was awarded a Fields medal for his contributions. Curiously, the classification features twenty-six exceptional examples—the sporadic simple groups—that do not belong to any of the natural families. It is natural to ask if they play a role in nature.

A sensational, yet partial answer to this question appeared just a few years later when McKay and Thompson noted coincidences[6] connecting the largest sporadic group—the Fischer–Griess monster[7], having about $8 \times 10^{53}$ elements—to the elliptic modular invariant $J(z)$, which first appeared a century earlier[8] in connection with the computation of lengths of arcs of ellipses. That sporadic groups and elliptic functions could be related seemed like lunacy. But Conway–Norton elaborated[9] on the observations of McKay and Thompson, and formulated the moonshine conjectures, which predict the existence of an infinite sequence of spaces $V_n^\natural$ for integers $n > 0$ that admit the monster group as symmetry in a systematic way. Nineteen other sporadic groups occur as building blocks of subgroups of the monster. Norton's generalised moonshine conjectures[10] formalised the notion that analogues of the $V_n^\natural$ should realise these.

Frenkel–Lepowsky–Meurman constructed[11] candidate spaces $V_n^\natural$, and the predictions of Conway–Norton were confirmed for them by Fields medal-winning work[12] of Borcherds in 1992. The notions of vertex algebra and Borcherds–Kac–Moody algebra arose from this, and now play fundamental roles in diverse fields of mathematics and physics. Consequently, the spaces $V_n^\natural$ may be recognised[13] as defining a bosonic string theory in 26 dimensions. The generalised moonshine conjectures were recently proven[14, 15] by Carnahan, so moonshine illuminates a physical origin for the monster, and for the 19 other sporadic groups that are involved in the monster. Therefore, 20 of the sporadic groups do indeed occur in nature.

But a unifying theory of the sporadic groups must also incorporate those six pariah sporadic groups that are not involved in the monster. The problem of uncovering moonshine for pariahs was posed in the seminal work[9] of Conway–Norton. Regarding moonshine from a different viewpoint, we have extended the theory[16] so as to incorporate two of these: the O'Nan group[17] and Janko's first group[18], the latter being a subgroup of the former. The elliptic modular invariant $J$ has the property that $J(z_Q)$ is a solution to a polynomial with integer coefficients whenever $z_Q = \frac{-B + i\sqrt{|D|}}{2A}$ for integers $A$, $B$ and $D$, with $D < 0$ and $D$ congruent to $B^2$ modulo $4A$. By suitably assembling these special values of $J$ we are led to a sequence of spaces $W_D$ for $D < 0$ that admit the O'Nan group as symmetry in a systematic way (see Theorem 1).

Using this we prove results on class numbers of quadratic forms (see Theorem 2), and Selmer groups and Tate–Shafarevich groups of elliptic curves (see Theorems 3 and 4). Thus we find that O'Nan moonshine sheds light on quantities that are central to current research in number theory. In particular, pariah groups of O'Nan and Janko do play a role in nature.

## Results

Our main results (Theorems 2, 3 and 4) reveal a role for the O'Nan pariah group as a provider of hidden symmetry to quadratic forms and elliptic curves. They also represent the intersection of moonshine theory with the Langlands program, which, since its inception in the 1960s, has become a driving force for research in number theory, geometry and mathematical physics.

**Hidden symmetry.** The Langlands program predicts an expansive system of hidden symmetry in algebraic statistics[19, 20]. For an example of this consider the Riemann zeta function

$$\zeta(s) := \prod_{p \text{ prime}} \frac{1}{1 - p^{-s}}. \tag{1}$$

This product converges for $s = \sigma + it$ a complex number with $\sigma > 1$, but at $s = 1$ equates to the harmonic series, and therefore diverges there. This is one proof that there are infinitely many primes. Setting $\Gamma(s) := \int_0^\infty v^{s-1} e^{-v} dv$ and $\theta(z) := \sum_{n=-\infty}^{\infty} e^{\pi i z n^2}$ we may write

$$\zeta(s) = \frac{\pi^{\frac{s}{2}}}{\Gamma\left(\frac{s}{2}\right)} \int_0^\infty v^{\frac{s}{2}} \left( \frac{\theta(iv) - 1}{2} \right) \frac{dv}{v}. \tag{2}$$

Riemann used (2) and the identity $v^{-\frac{1}{2}} \theta(iv^{-1}) = \theta(iv)$ to show[21] that $\zeta(s)$ can be defined for all complex numbers except $s = 1$, in such a way that the completed zeta function $\xi(s) := \pi^{-\frac{s}{2}} \Gamma\left(\frac{s}{2}\right) \zeta(s)$ is invariant under the two-fold symmetry that swaps $s$ with $1 - s$.

For the Langlands program $\zeta(s)$ is the first in a family arising from Diophantine analysis, which is the classical art[22] of finding rational solutions to polynomials with integer coefficients. Linear equations are controlled by $\zeta(s)$, and the next examples relate quadratic forms $Q(x, y) := Ax^2 + Bxy + Cy^2$ to the Dirichlet $L$-series

$$L_D(s) := \prod_{p \text{ prime}} \frac{1}{1 - \chi_D(p) p^{-s}}. \tag{3}$$

Here $A$, $B$ and $C$ are integers, $D := B^2 - 4AC$ is the discriminant of $Q$, and $\chi_D(p)$ is a certain function depending on $D$. If $D = 1$ then $L_D(s) = \zeta(s)$. Dirichlet $L$-series admit completions which have the same two-fold symmetry as $\zeta(s)$. The as yet unresolved generalised Riemann hypothesis[23] predicts that if $L_D(s) = 0$ for $s = \sigma + it$ with $0 < \sigma < 1$ then $\sigma = \frac{1}{2}$.

Just as for $\zeta(s)$, the behaviour of $L_D(s)$ near $s = 1$ is important. Call a discriminant $D$ fundamental if it cannot be written as $d^2 D'$ where $d$ is an integer greater than 1 and $D'$ is the discriminant of another quadratic form. If we focus on the values $Q(x, y)$ for $x$ and $y$ integers then it is the same to consider $Q'(x, y) := Q(ax + by, cx + dy)$, for any integers $a$, $b$, $c$ and $d$, so long as $ad - bc = 1$. Such a modular transformation preserves discriminants, so we may consider the forms with given discriminant $D$, and count them modulo modular transformations. For $D$ fundamental this is the class number $h(D)$. For example, $h(-7) = 1$ because $Q(x, y) := x^2 + xy + 2y^2$ has discriminant $-7$, and if $Q'(x, y)$ is another such form then $Q'(x, y) = Q(ax + by, cx + dy)$ for some integers $a$, $b$, $c$ and $d$ with $ad - bc = 1$. Gauss

conjectured[24] that $h(D)$ takes any given value only finitely many times for negative $D$. Dirichlet proved[25]

$$\frac{\sqrt{|D|}}{2\pi}L_D(1) = \frac{h(D)}{2} \qquad (4)$$

for fundamental $D < -4$. Using this, Siegel solved the Gauss conjecture in a weak sense in 1944 by showing[26] that for any $\epsilon > 0$ there is a constant $c$ such that $h(D) > c|D|^{\frac{1}{2}-\epsilon}$ for $D < 0$. But the method is not effective because the constant $c$ it produces depends upon the validity of the generalised Riemann hypothesis. The best known effective lower bound[27, 28] takes the form $h(D) > c(\log|D|)^{1-\epsilon}$ for $D < 0$.

**Elliptic curves**. It is a familiar fact that $3^2 + 4^2 = 5^2$. Fermat's last theorem claims that if $x$, $y$ and $z$ are integers such that $x^n + y^n = z^n$ with $n > 2$ then at least one of them is zero. Wiles famously proved this[29] by establishing hidden symmetry in the Diophantine analysis of

$$y^2 = x^3 + Ax + B. \qquad (5)$$

If $4A^3 + 27B^2 \neq 0$ then this cubic equation defines an elliptic curve $E$, an integer $N$ called the conductor of $E$, and a Hasse–Weil $L$-series

$$L_E(s) := \prod_{p \text{ prime}} \frac{1}{1 - a_p p^{-s} + \varepsilon(p)p^{1-2s}}. \qquad (6)$$

Here $a_p$ is an integer depending on $E$ and $p$, and $\varepsilon(p)$ is 0 or 1 according as $p$ divides $N$ or not. Modularity for $E$ is the existence[30] of a modular form $f_E(z)$, (complex) differentiable for $z = u + iv$ with $v > 0$, such that $(Ncz + d)^{-2}f_E\left(\frac{az+b}{Ncz+d}\right) = f_E(z)$ for any integers $a$, $b$, $c$ and $d$ with $ad - Nbc = 1$, and

$$L_E(s) = \frac{(2\pi)^s}{\Gamma(s)} \int_0^\infty v^s f_E(iv) \frac{dv}{v}. \qquad (7)$$

For $L_E(s)$ the behaviour near $s = 1$ is again important, but not yet understood, and a focus of current research. The Birch–Swinnerton–Dyer (BSD) conjecture predicts[31, 32] that it is a key that unlocks the rational solutions to the equation defining $E$. Specifically, if $r_E$ is the rank of $E$, representing the number of independent infinite families of rational solutions, then $\lim_{s \to 1}(s - 1)^{-r_E}L_E(s)$ should be finite and non-zero. In particular, $L_E(1)$ should vanish if and only if there are infinitely many rational solutions. The Tate–Shafarevich group Ш $(E)$ measures the extent to which computations with $E$ can be carried out modulo primes. A stronger form of the BSD conjecture asserts that

$$\lim_{s \to 1} \frac{1}{\Omega_E} \frac{L_E(s)}{(s-1)^{r_E}} = c_E |\text{Ш}(E)| \qquad (8)$$

for certain computable constants $\Omega_E$ and $c_E$, where $|\text{Ш}(E)|$ is the cardinality of Ш$(E)$.

The BSD conjecture is known only in its weak form[28, 33] for $r_E = 0$ and $r_E = 1$. The strong form gives us complete control once we know Ш$(E)$ and $r_E$. For each prime $p$ there is a Selmer group $\text{Sel}_p(E)$, which ties together the $p$-fold symmetries in Ш$(E)$ with the infinite families of rational solutions to $E$, so Selmer groups and Tate–Shafarevich groups are of primary importance.

The moonshine that we establish for the O'Nan group (see Theorem 1) enables us to prove new constraints on class numbers $h(D)$ (see Theorem 2), and empowers us to relate certain Selmer groups $\text{Sel}_p(E)$ and Tate–Shafarevich groups Ш$(E)$ to effectively computable series $a_g(D)$ (see Theorems 3 and 4).

**Moonshine**. The elliptic modular invariant $J$ is the unique (complex) differentiable function of $z = u + iv$ for $v > 0$ that satisfies $J\left(\frac{az+b}{cz+d}\right) = J(z)$ when $a$, $b$, $c$ and $d$ are integers such that $ad - bc = 1$, and $\lim_{v \to \infty}(J(iv) - e^{2\pi v}) = 0$. The connection between the monster and $J$ is that the dimension of $V_n^\natural$ is the coefficient of $e^{2n\pi iz}$ in the Fourier expansion

$$J(z) = e^{-2\pi iz} + 196884e^{2\pi iz} + 21493760e^{4\pi iz} + 864299970e^{6\pi iz} + \ldots \qquad (9)$$

Recursion relations[34] compute these Fourier coefficients effectively.

Let $F(z)$ be the unique (complex) differentiable function of $z = u + iv$ for $v > 0$ such that

1. $(4cz + d)^{-2}\tilde{F}\left(\frac{az+b}{4cz+d}\right) = \tilde{F}(z)$ when $\tilde{F}(z) := F(z)\theta(2z)$ and $a$, $b$, $c$ and $d$ are integers satisfying $ad - 4bc = 1$,

2. $F^+\left(z + \frac{1}{4}\right) = F^+(z)$   and   $F^-\left(z + \frac{1}{4}\right) = -iF^-(z)$   for $F^\pm(z) := \frac{1}{2}\left(F(z) \pm F\left(z + \frac{1}{2}\right)\right)$,

3. $\lim_{v \to \infty}(F(iv) + e^{8\pi v})$ is finite.

Under these conditions $F$ is related to $J$ by $F\left(z + \frac{1}{2}\right)\theta(8z) + F^-(z)\theta(2z) = \frac{1}{8\pi i}\frac{d}{dz}J(4z)$, and recursion relations[35] compute the Fourier coefficients of $F$ effectively.

**Theorem 1**. There exists a sequence of spaces $W_D$ for discriminants $D < 0$ that admit the O'Nan group as symmetry, the dimension of $W_D$ being the coefficient of $e^{2|D|\pi iz}$ in the Fourier expansion

$$F(z) = -e^{-8\pi iz} + 2 + 26752e^{6\pi iz} + 143376e^{8\pi iz} + 8288256e^{14\pi iz} + \ldots \qquad (10)$$

We sketch the proof of Theorem 1. Let $a(D)$ denote the coefficient of $e^{2|D|\pi iz}$ in the Fourier expansion of $F$, so that we have $F(z) := -e^{-8\pi iz} + 2 + \sum_{D<0}a(D)e^{2|D|\pi iz}$. Theorem 1 amounts to the association of $a(D) \times a(D)$ matrices to symmetries in the O'Nan group, for each $D < 0$. Taking $a_g(D)$ to be the trace (i.e., sum of diagonal entries) of the matrix corresponding to a symmetry $g$ we obtain a new function $F_g(z) := -e^{-8\pi iz} + 2 + \sum_{D<0}a_g(D)e^{2|D|\pi iz}$. The O'Nan group has a character table[17], which encodes the possible traces $a_g(D)$ that can arise, and thereby equips the $F_g$ with special properties. Conversely, to specify functions $F_g$ that are compatible with the character table is enough to prove that corresponding matrices exist, and enough to confirm moonshine for the O'Nan group. This is how we prove Theorem 1. We first realise the $F_g$ as modular forms satisfying conditions analogous to those defining $F$. Then we use those conditions to prove growth estimates and congruences for the $a_g(D)$. For example, if $g$ is a seven-fold symmetry then it develops that $a(D)$ is congruent to $a_g(D)$ modulo 343, for every $D < 0$. Finally, we use the growth estimates and congruences to prove compatibility with the character table.

Define $J_{O'N}(z) := J(z)^2 - J(z) - 393768$. For $Q(x, y) = Ax^2 + Bxy + Cy^2$ a quadratic form with $D = B^2 - 4AC < 0$ set $z_Q := \frac{-B + i\sqrt{|D|}}{2A}$. Set $w_Q := 6$ if $Q$ is equivalent (i.e. related by a modular transformation) to $A'x^2 + A'xy + A'y^2$ for some $A'$, set $w_Q := 4$ if $Q$ is equivalent to $A'x^2 + A'y^2$, and set $w_Q := 2$ otherwise. Then for $D < 0$ we have[35]

$$a(D) = \sum_{[B^2 - 4AC = D]} \frac{J_{O'N}(z_Q)}{w_Q}, \qquad (11)$$

where the sum is over equivalence classes of quadratic forms of discriminant $D$. If $D$ is fundamental then $h(D)$ is the number of summands in (11). For example, $\dim W_{-7} = \frac{1}{2} J_{O'N}\left(\frac{-1+i\sqrt{7}}{2}\right) = 8288256$ because every quadratic form with discriminant $-7$ is equivalent to $x^2 + xy + 2y^2$. So the identity (11) hints at a connection to class numbers. Say that $D$ is not a square modulo $p$ if there is no integer $n$ such that $D$ is congruent to $n^2$ modulo $p$. By establishing analogues of (11) for $a_g(D)$, for $g$ a two-fold, three-fold, five-fold or seven-fold symmetry in the O'Nan group, we obtain the following results.

**Theorem 2**. Let $D < 0$ be a fundamental discriminant. If $D < -8$ is even then $-24h(D)$ is congruent to $a(D)$ modulo 16. If $D$ is not a square modulo 3 then $-24h(D)$ is congruent to $a(D)$ modulo 9. If $p$ is 5 or 7, and if $D$ is not a square modulo $p$ then $-24h(D)$ is congruent to $a(D)$ modulo p.

For $D < 0$ a fundamental discriminant define elliptic curves $E_{14}(D)$ and $E_{15}(D)$ as follows.

$$E_{14}(D): y^2 = x^3 + 5805D^2x - 285714D^3$$
$$E_{15}(D):: y^2 = x^3 - 12987D^2x - 263466D^3 \tag{12}$$

By proving analogues of (11) for higher order symmetries in the O'Nan group we obtain new relationships between class numbers and the rational solutions to these equations.

**Theorem 3**. Let $D < 0$ be a fundamental discriminant. Suppose that $D$ is congruent to 1 modulo 2 and is not a square modulo 7, and let $g$ be a two-fold symmetry in the O'Nan group. Then $\mathrm{Sel}_7(E_{14}(D))$ is non-trivial if and only if $a_g(D)$ is congruent to $3h(D) - 9h^{(2)}(D)$ modulo 7. Also, if $L_{E_{14}(D)}(1) \neq 0$ then the Birch–Swinnerton-Dyer conjecture is true for $E_{14}(D)$, and $|\text{Ш}(E_{14}(D))|$ is congruent to 0 modulo 7 if and only if $a_g(D)$ is congruent to $3h(D) - 9h^{(2)}(D)$ modulo 7.

**Theorem 4**. Let $D < 0$ be a fundamental discriminant. Suppose that $D$ is congruent to 1 modulo 3 and is not a square modulo 5, and let g be a three-fold symmetry in the O'Nan group. Then $\mathrm{Sel}_5(E_{15}(D))$ is non-trivial if and only if $a_g(D)$ is congruent to $2h(D) - 4h^{(3)}(D)$ modulo 5. Also, If $L_{E_{15}(D)}(1) \neq 0$ then the Birch–Swinnerton-Dyer conjecture is true for $E_{15}(D)$, and $\text{Ш}(E_{15}(D))|$ is congruent to 0 modulo 5 if and only if $a_{\,g}(D)$ is congruent to $2h(D) - 4h^{(3)}(D)$ modulo 5.

Note that $a_g(D)$ is the same for all two-fold symmetries $g$, and this is also true for three-fold symmetries. Similar to the $a(D)$, the $a_g(D)$ are computed effectively by recursion relations when $g$ is a two-fold or three-fold symmetry. The $h^{(p)}(D)$ are analogues of the class numbers $h(D)$ that are defined by restricting to modular transformations $Q(ax + by, pcx + dy)$ where $ad - pbc = 1$.

Recent work of Skinner[36], following earlier work[37] of Skinner–Urban, shows that (8) holds modulo certain primes $p$, when certain conditions on the underlying elliptic curve $E$ are satisfied. We prove Theorems 3 and 4 by verifying these conditions for $p = 2$ and $E = E_{14}(D)$, and for $p = 3$ and $E = E_{15}(D)$, when $D < 0$ is fundamental.

Define a further two families of elliptic curves as follows.

$$E_{11}(D): y^2 = x^3 - 13392D^2x - 1080432D^3$$
$$E_{19}(D): y^2 = x^3 - 12096D^2x - 544752D^3 \tag{13}$$

If we assume (8) then the analogues of (11) for eleven-fold and nineteen-fold symmetries in the O'Nan group lead to the statement that if $p = 11$ or $p = 19$, and $D < 0$ is a fundamental discriminant that is not a square modulo $p$, then $\mathrm{Sel}_p(E_p(D))$ is non-trivial if and only if $a(D)$ is congruent to $-24h(D)$ modulo $p$.

## Discussion

Moonshine has had a powerful impact on mathematics and theoretical physics. For instance, the notions of vertex algebra and Borcherds–Kac–Moody algebra—discovered en route to the positive resolution of the moonshine conjectures by Borcherds—are now fields in their own right, with applications in string theory and the geometric counterpart to the Langlands program. The extension of moonshine to pariah groups opens the door to exciting directions for future research. For one, it is natural to ask if there is analogous moonshine for the remaining four pariah groups. Preliminary evidence suggests that the answer to this question is positive. It is also natural to ask for a fuller understanding of the hidden symmetry that pariah groups, and perhaps also other finite simple groups, provide to problems in Diophantine analysis. Moving beyond elliptic curves, there are thirty-one-fold symmetries in the O'Nan group that can be used to prove congruences that control rational solutions to a certain family of quintic equations[16]. A third natural question concerns a physical origin for the pariahs. It is natural to ask if there is a string theoretic construction of the spaces $W_D$ of O'Nan moonshine analogous to that found for the spaces $V_n^\natural$ by Frenkel–Lepowsky–Meurman.

**Data availability**. Data sharing is not applicable to this article, as no data sets were generated or analysed during the current study.

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

## Acknowledgements

This research was supported by the Asa Griggs Candler Fund (K.O.), the Max-Planck-Institut für Mathematik in Bonn (M.H.M.), the U.S. National Science Foundation, DMS 1601306 (J.F.R.D. and K.O.), and the Simons Foundation, #316779 (J.D.).

## Author contributions

J.F.R.D., M.H.M. and K.O. performed the research and wrote the manuscript.

## Additional information

**Competing interests:** The authors declare no competing financial interests.

