## [Peer Review File · Nature Communications]

REVIEWERS' COMMENTS:

Reviewer #2 (Remarks to the Author):

Referee report on duncan-mertens-ono (due June 17)

This paper marks a new direction for the field of moonshine, for multiple reasons. First, it is the first example where one has exceptional behavior attached to sporadic groups outside the "happy family" of monster subquotients. Second, the authors find connections between congruences of McKay-Thompson series and the Selmer and Tate-Shafarevich groups of certain optimal elliptic curves.

The main claims are:

- 1) there exists a virtual graded O'Nan module whose graded characters are mock modular forms of weight $3/2$. In fact, they are modular forms except for one case attached to a class of order 16.
- 2) for small primes dividing the order of O'Nan, we get congruences between graded characters of this virtual module and class numbers.
- 3) for $p=11$ and 19 , congruences are equivalent to nontriviality of p -torsion in the Selmer group or Tate-Shafarevich group of a quadratic twist of some optimal elliptic curve (the latter under a non-vanishing L-value condition).
- 4) For 5 and 7 , congruences are equivalent to a similar non-triviality, but the elliptic curves are chosen slightly differently.

These are strikingly original results, and I strongly recommend publication.

Comments on article

Page 2, first line: The ordering of "molecules" and "atoms" should be reversed to make the analogy with simple groups and finite symmetries flow better. For example, "As atoms constitute molecules...".

Page 3, middle: I think it is a stretch to say that "twists" of the graded pieces of the moonshine module are what were conjectured in generalized moonshine. Perhaps something weaker, like "analogous spaces" would be more appropriate.

Page 8 last paragraph: As far as I can tell, known connections between modern encryption and BSD are tenuous at best. The claim that BSD impacts our everyday lives needs to be substantially weakened.

Page 12 second from last line: "lthe" should be "the"

Page 12-13: Should theorem 4 have an assertion about validity of BSD for $E_{15}(D)$, like Theorem 3 has for $E_{14}(D)$? It looks like both follow from Theorem 1.4 in the ArXiv preprint.

Page 17: Reference 14 should include "ArXiv e-prints" for uniformity.

Page 18: Reference 21 should start with "Über" instead of Ueber. The former is the title attached to the paper as published, and the latter is the title attached to the paper in Riemann's collected works.

Comments on manuscript/ArXiv paper

Page 6: I don't see why the second remark after Theorem 1.2 exists - it seems to follow immediately from the claims of the theorem.

Reviewer #3 (Remarks to the Author):

This well-written paper describes the most convincing appearance known to me of one (or two!) of the pariahs in mathematics removed from group theory. Only time will tell how significant these observations are. The biggest question this work begs, is whether there is a physical interpretation of this work, via string theory.

An instance of moonshine is an unexpected juxtaposition of an infinite-dimensional \mathbb{Z} -graded representation of a finite group, with some flavour of modular forms, and ideally with a connection to physics (string theory or conformal field theory) or geometry. There should be some sort of uniqueness emphasizing the singularity of this juxtaposition. The most famous example is the monstrous moonshine of Conway-Norton, but we are accumulating more and more examples in recent years, in particular the Mathieu moonshine of Eguchi-Ooguri-Tachikawa, and its generalization umbral moonshine by Cheng-Duncan-Harvey. The importance of monstrous moonshine is now beyond question. The importance or relevance of Mathieu and umbral moonshines is still uncertain, in my opinion.

The authors introduce a new moonshine, more number-theoretic than the others. What is especially interesting about it is that it concerns groups unrelated to the monster. The relation of O'Nan - a group over 1000 times larger than the groups appearing in Mathieu and umbral moonshines - to class numbers of quadratic forms and e.g. Tate-Shafarevich groups of elliptic curves, is every bit as surprising to me as the relation of the monster to the j -function.

The exposition is creatively written, in such a way that it will be more accessible to a wide scientific audience. I think this article will be interesting to a wide range of readers. Certainly there is no question of the validity of their claims. I would have a hard time arguing though that this work is "of extreme importance to scientists in the specific field", but there are very few if any works in mathematics I would describe in that way. Surely, importance in mathematics takes years to decide. This discovery is certainly quite unexpected though.

Some comments/questions:

(i) On page 2, the authors write that "by the 1950s it was expected that most, perhaps all, finite simple groups belonged to infinite families. Surely by then it was well appreciated that the five Mathieu groups existed and were simple. I think "perhaps all" should be deleted.

(ii) On page 3, we read that 1976 was the year McKay and Thompson noted their coincidences. I thought it was 1978.

(iii) It is a little glib on page 5 to say that the pole of the Riemann zeta at $s=1$ is "because there are infinitely many primes". If the primes p_n are replaced by other infinite sequences (e.g. their squares p_n^2), then $\prod_n 1/(1-p_n^{-1})$ could converge. A more accurate explanation, which all readers could appreciate, is that the divergence of that product at $s=1$ is because the harmonic series $\sum_n n^{-1}$ diverges.

(iv) In Theorem 3, there is a typo "ithe".

- Page 2, first line: The ordering of "molecules" and "atoms" should be reversed to make the analogy with simple groups and finite symmetries flow better. For example, "As atoms constitute molecules...". **CHANGE MADE**
- Page 3, middle: I think it is a stretch to say that "twists" of the graded pieces of the moonshine module are what were conjectured in generalized moonshine. Perhaps something weaker, like "analogous spaces" would be more appropriate. **CHANGE MADE USING SUGGESTED LANGUAGE**
- Page 8 last paragraph: As far as I can tell, known connections between modern encryption and BSD are tenuous at best. The claim that BSD impacts our everyday lives needs to be substantially weakened. **STATEMENT REMOVED**
- Page 12 second from last line: "Ithe" should be "the" **TYPO CORRECTED**
- Page 12-13: Should theorem 4 have an assertion about validity of BSD for $E_{15}(D)$, like Theorem 3 has for $E_{14}(D)$? It looks like both follow from Theorem 1.4 in the ArXiv preprint. **CHANGE MADE**
- Page 17: Reference 14 should include "ArXiv e-prints" for uniformity. **CHANGE MADE**
- Page 18: Reference 21 should start with Über instead of Ueber. The former is the title attached to the paper as published, and the latter is the title attached to the paper in Riemann's collected works. **TYPO CORRECTED**

Referee 3.

- (i) On page 2, the authors write that "by the 1950s it was expected that most, perhaps all, finite simple groups belonged to infinite families. Surely by then it was well appreciated that the five Mathieu groups existed and were simple. I think "perhaps all" should be deleted. **CHANGE MADE**
- (ii) On page 3, we read that 1976 was the year McKay and Thompson noted their coincidences. I thought it was 1978. **CHANGE MADE.**
- (iii) It is a little glib on page 5 to say that the pole of the Riemann zeta at $s=1$ is "because there are infinitely many primes". If the primes p_n are replaced by other infinite sequences (e.g. their squares p^2), then $\prod_n 1/(1 - p_n^{-1})$ could converge. A more accurate explanation, which all readers could appreciate, is that the divergence of that product at $s=1$ is because the harmonic series $\sum_n n^{-1}$ diverges. **AGREED. LANGUAGE MODIFIED**
- (iv) In Theorem 3, there is a typo "Ithe". **TYPO CORRECTED**